# Nicotinamide Mononucleotide Prevents Free Fatty Acid-Induced Reduction in Glucose Tolerance by Decreasing Insulin Clearance

**DOI:** 10.3390/ijms222413224

**Published:** 2021-12-08

**Authors:** Ashraf Nahle, Yemisi Deborah Joseph, Sandra Pereira, Yusaku Mori, Frankie Poon, Hilda E. Ghadieh, Aleksandar Ivovic, Tejas Desai, Simona S. Ghanem, Suman Asalla, Harrison T. Muturi, Emelien M. Jentz, Jamie W. Joseph, Sonia M. Najjar, Adria Giacca

**Affiliations:** 1Department of Physiology, Faculty of Medicine, University of Toronto, Toronto, ON M5S 1A8, Canada; ash.nahle@mail.utoronto.ca (A.N.); deborah.joseph@mail.utoronto.ca (Y.D.J.); sandra.pereira@alum.utoronto.ca (S.P.); u-mori@med.showa-u.ac.jp (Y.M.); frankie.poon@mail.utoronto.ca (F.P.); a.ivovic@mail.utoronto.ca (A.I.); tejas.desai@mail.utoronto.ca (T.D.); 2Division of Diabetes, Metabolism and Endocrinology, Showa University School of Medicine, Shinagawa, Tokyo 142-0064, Japan; 3Center for Diabetes and Endocrine Research, College of Medicine and Life Sciences, University of Toledo, Toledo, OH 43606, USA; hg36@aub.edu.lb (H.E.G.); Simona.Ghanem@rockets.utoledo.edu (S.S.G.); najjar@ohio.edu (S.M.N.); 4Department of Biomedical Sciences, Heritage College of Osteopathic Medicine, Ohio University, Athens, OH 45701, USA; asalla@ohio.edu (S.A.); muturi@ohio.edu (H.T.M.); 5School of Pharmacy, University of Waterloo, Kitchener, ON N2G 1C5, Canada; emjentz@uwaterloo.ca (E.M.J.); jamie.joseph@uwaterloo.ca (J.W.J.); 6Diabetes Institute, Heritage College of Osteopathic Medicine, Ohio University, Athens, OH 45701, USA; 7Banting and Best Diabetes Centre, University of Toronto, Toronto, ON M5G 2C4, Canada; 8Department of Medicine, University of Toronto, Toronto, ON M5S 1A1, Canada; 9Institute of Medical Science, University of Toronto, Toronto, ON M5S 1A8, Canada

**Keywords:** type 2 diabetes, insulin resistance, insulin clearance, SIRTs, glucose tolerance, free fatty acids

## Abstract

The NAD-dependent deacetylase SIRT1 improves β cell function. Accordingly, nicotinamide mononucleotide (NMN), the product of the rate-limiting step in NAD synthesis, prevents β cell dysfunction and glucose intolerance in mice fed a high-fat diet. The current study was performed to assess the effects of NMN on β cell dysfunction and glucose intolerance that are caused specifically by increased circulating free fatty acids (FFAs). NMN was intravenously infused, with or without oleate, in C57BL/6J mice over a 48-h-period to elevate intracellular NAD levels and consequently increase SIRT1 activity. Administration of NMN in the context of elevated plasma FFA levels considerably improved glucose tolerance. This was due not only to partial protection from FFA-induced β cell dysfunction but also, unexpectedly, to a significant decrease in insulin clearance. However, in conditions of normal FFA levels, NMN impaired glucose tolerance due to decreased β cell function. The presence of this dual action of NMN suggests caution in its proposed therapeutic use in humans.

## 1. Introduction

Type 2 diabetes mellitus (T2DM) is a growing health concern worldwide, in parallel to increased incidences of obesity. It is associated with insulin resistance in combination with decreased secretory pancreatic β cell function (i.e., decreased ability of β cells to compensate for insulin resistance). In overweight and obese individuals, who represent ~90% of T2DM patients [1], insulin resistance is partly caused by elevated levels of circulating free fatty acids (FFAs).

In addition to increased insulin secretion, decreased insulin clearance also results in elevation of circulating insulin levels to contribute to the compensation for insulin resistance. Secreted insulin is cleared mainly by hepatocytes through endocytosis of the insulin-insulin receptor complex [2] and, to a lesser extent, by the kidney [3].

SIRT1 is an NAD-dependent protein deacetylase that improves glucose tolerance by increasing insulin secretion and sensitivity [4]. Our laboratory investigated the protective effect of SIRT1 on β cells in the context of high plasma FFA levels in vivo. We found that intravenous co-infusion of resveratrol (a SIRT1 activator [5]) partially protected against β cell dysfunction induced by a 48 h intravenous oleate infusion. We also found that β cell-specific sirtuin 1-overexpressing (BESTO) transgenic mice are partially protected against oleate-induced β cell dysfunction in vivo [6].

C57BL/6J mice fed a high-fat diet displayed impairment of NAD biosynthesis, decreased SIRT1 activity, and diabetes [7]. Administration of NMN, the rate-limiting intermediate in NAD biosynthesis, restored normal tissue NAD levels and increased glucose-stimulated insulin secretion (GSIS) and glucose tolerance in diet-induced diabetes in mice [7]. NMN is also being investigated as a therapeutic modality against diabetes-related disorders in multiple clinical trials [8,9,10].

NMN administration avoids the non-specific antioxidant effects, which are characteristic of resveratrol infusion, as well as the high, non-physiological levels of SIRT1 expression exhibited by the BESTO model used in our previous study [6]. Based on this study and on the adverse effects of a high-fat diet on NAD biosynthesis and SIRT1 activity [7], we hypothesized that increased SIRT1 activity by NMN administration counters oleate-induced β cell dysfunction in vivo. The current studies were carried out to test this hypothesis. To this end, we infused oleate (OLE) or saline (SAL) with or without NMN intravenously for 48 h before performing a hyperglycemic clamp to evaluate β cell function and overall glucose and insulin metabolism.

## 2. Results

At the end of the 48-h IV infusion prior to the hyperglycemic clamp (time −20 to 0 min), glycemia was slightly lower in the NMN + OLE group, compared with the SAL and NMN groups, and in the OLE group, compared with the NMN group (Figure 1A). Starting from time = 0, blood glucose levels were elevated by glucose infusion to a steady-state level that was similar between all groups (~22 mM). There was no significant difference in glycemia among the groups during the last 30 min of the hyperglycemic clamp. However, the average glycemic levels in the NMN + OLE group during the entire clamp period (0–120 min) were lower than those of the individual OLE and NMN groups. This occurred despite the fact that the NMN + OLE group was intravenously infused with glucose at a significantly higher rate than all of the other groups (Figure 1B).

The glucose infusion rate (GINF) required to maintain hyperglycemia is a reflection of glucose tolerance. As expected, the GINF required to maintain steady-state hyperglycemia during the last 30 min of the clamp was significantly lower in the OLE- than the SAL-treated group (Figure 1B). Remarkably, mice co-infused with NMN + OLE had a substantial increase in GINF, compared with the OLE and the SAL groups. Interestingly, infusing NMN in the absence of elevated FFA resulted in a significant decrease in GINF relative to SAL infusion (Figure 1B).

Plasma insulin levels were similar between the SAL, OLE, and NMN groups prior to intravenous glucose infusion and during the hyperglycemic clamp (Figure 1C). At basal glycemia, the plasma concentration of insulin was significantly higher in the NMN + OLE group, compared with the SAL group (Figure 1C). Under hyperglycemic clamp conditions, plasma insulin levels rose in the NMN + OLE group by ≥2-fold relative to the other groups (Figure 1C). At basal glycemia, plasma C-peptide levels, although not statistically different, tended to be lower in the OLE and NMN + OLE groups relative to the SAL group. (Figure 1D). During the hyperglycemic clamp, the NMN + OLE group manifested significantly greater C-peptide levels, compared with the OLE and NMN groups. However, this increase in C-peptide levels was not proportional to the increase in insulin levels in the NMN + OLE group, compared with the other groups.

During basal glycemia, OLE infusion did not significantly decrease the Insulin Clearance Index ([C-peptide]/[Insulin]), compared with SAL. In contrast, during clamped hyperglycemia, OLE infusion resulted in a significant decrease in insulin clearance, compared with SAL infusion (Figure 2A). Interestingly, the co-infusion of NMN + OLE resulted in a significant decrease in insulin clearance during basal glycemia and hyperglycemia, compared with SAL infusion. This demonstrated that NMN amplified the oleate-induced decrease in insulin clearance. As expected, oleate infusion resulted in a significant decrease in the Insulin Sensitivity Index (M/I), compared with SAL infusion (Figure 2B). NMN infusion with or without OLE did not significantly alter insulin sensitivity, compared with the OLE and SAL groups. This is unlike our previous studies with resveratrol which prevented fat-induced insulin resistance [11].

As expected from our previous studies [12,13], OLE infusion significantly decreased β cell function, i.e., the Disposition Index (DI), compared with SAL infusion (Figure 2C). NMN + OLE co-infusion did not significantly alter the DI relative to SAL and OLE individual infusions. Hence, NMN + OLE co-infusion resulted in an intermediate DI, suggesting partial protection from FFA-induced β cell dysfunction. The hypothesis that β cell function is improved by NMN in the context of elevated plasma levels of FFA is supported by the significantly higher C-peptide levels (indicator of absolute insulin secretion) in the NMN + OLE group, compared with the OLE group. Mice infused with NMN in the absence of elevated FFA levels exhibited a significantly lower DI, compared with the SAL group (Figure 2C).

OLE infusion caused a 1.5 to 2-fold elevation of plasma FFA levels, compared with SAL infusion (Figure 2D). Consistent with previous findings [7], co-infusion of NMN significantly lowered plasma FFA levels, compared with OLE alone, and individual NMN infusion had a similar trend compared with SAL. At the end of the hyperglycemic clamp, plasma FFA levels in all groups were moderately lower than their pre-clamp levels, owing to hyperinsulinemia-driven increase in FFA esterification and reduced lipolysis by adipocytes. Consistently, the NMN + OLE group showed the greatest decrease in plasma FFA levels (Figure 2D) in parallel to its higher plasma insulin levels than the other groups (Figure 1C). NMN infusion caused an increase in hepatic NAD+ levels NMN + OLE infusion increased NAD+/NADH ratio, partly due to a decrease in NADH levels (Appendix A).

Since we unexpectedly found a decrease in insulin clearance in the NMN + OLE versus the OLE group, and in light of our previous finding that FFAs decrease the expression of CEACAM1 [14] that plays a key role in hepatic insulin clearance [14,15], we then assessed hepatic CEACAM1 protein levels by Western blot analysis. As depicted in Figure 3, CEACAM1 protein expression was significantly lower in the livers of mice treated with NMN + OLE than those treated with SAL.

To investigate the mechanism(s) underlying the reduction in CEACAM1, we overexpressed SIRT1 in HepG2 human hepatoma cells via adenoviral-mediated delivery. As Table 1 shows, this did not alter CEACAM1 mRNA levels. Because NMN activates other SIRTs, such as SIRT2 that activates FOXO-1 [16], which, in turn, transcriptionally represses CEACAM1 expression [17], we next investigated the effect of adenoviral-mediated overexpression of SIRT2 in HepG2. Unexpectedly, SIRT2 overexpression increased CEACAM1 mRNA levels by 50%), whereas the SIRT inhibitor sirtinol (predominantly a SIRT2 inhibitor) markedly decreased CEACAM1 mRNA levels (Table 1).

We then tested whether SIRT2 inhibitors might have been produced in vivo by NMN metabolism to, in turn, reduce CEACAM1 expression. In concert with this hypothesis, the serum concentration of the pan-SIRT inhibitor nicotinamide was increased by two-fold following 48 h intravenous infusion of NMN (Figure 4A).

Because CEACAM1 and insulin clearance were decreased in mice only when NMN was co-infused with oleate, but not when NMN was infused individually, we further hypothesized that oleate may have accentuated the inhibitory effect of nicotinamide on SIRT2, which in turn would have resulted in decreased CEACAM1 and insulin clearance. However, the liver expression of SIRT2 was reduced by NMN but not by NMN + OLE, whereas SIRT1 expression was unchanged (Figure 5A,B).

FFA effect to decrease Ceacam1 transcription is PPARα-mediated [18], therefore we assessed markers of PPARα activity. We found that the mRNA level of Cpt1, a transcriptional target of PPARα, was increased by oleate, irrespective of NMN (Figure 6C), but other PPARα target genes, such as Acox1 and Aco1 appeared to be upregulated by NMN (Figure 5D,E), in accordance with increased SIRT1 activity deacetylating PGC-1α. We initially hypothesized that, similarly, NMN may have accentuated the effect of PPARα on Ceacam1 transcription via PGC-1α activation by SIRT1. However, we did not observe any reduction in insulin clearance when the SIRT1 activator resveratrol was co- infused with oleate in rats [6], Furthermore, as described above, SIRT1 overexpression did not decrease Ceacam1 mRNA in HepG2 cells whereas sirtinol, a SIRT2 inhibitor, did.

A possible synergism between low SIRT2 and oleate is at the level of oxidative stress. SIRT2 is known to reduce oxidative stress [19], although it has also been reported to increase it via activating JNK by deacetylation [20]. JNK can increase CEACAM1 transcription via AP-1 [21]. Therefore, we assessed JNK activation in our liver samples. However, we did not find any significant difference in JNK phosphorylation (marker of activation) among groups (Figure 4B).

We have previously shown that infusing Intralipid, a triglyceride emulsion, for 7 h induced oxidative stress and decreased CEACAM1 levels in rats, and that this effect was prevented by treatment with the anti-oxidant N-acetyl-L-cysteine [14]. In the present study, Nox1 and Nox4, which are oxidative stress inducing genes, were upregulated to a greater extent in the NMN + OLE group relative to the other groups (Figure 5F,G). Thus, oleate and low SIRT2 could have synergized at the level of oxidative stress to trigger repression of CEACAM1 transcription in several ways. Oxidative stress could impair insulin action, which would in turn downregulate CEACAM1 transcription [14,21]. In our previous study, 7 h infusion of Intralipid increased serine phosphorylation of IRS in the liver, a marker of hepatic insulin resistance [14]. In the present study, serine phosphorylation of IRS did not differ among groups (Figure 4C), in accordance with our previous studies where 48 h infusion of Intralipid showed hepatic insulin resistance despite unchanged IRS serine phosphorylation [22]. In addition to causing insulin resistance, oxidative stress could induce HIF-1α and HIF-2α [23], as shown by the preferential increase in the mRNA levels of these transcription factors in the NMN + OLE group (Figure 5H,I). HIFs, in particular HIF-2α, could downregulate CEACAM1 expression in hepatocytes [24].

Since hyperinsulinemia and reduced hepatic CEACAM1 can induce hepatic steatosis, thereby potentially amplifying the effect of oleate [25,26,27], we assessed hepatic fat content using oil-red-O staining. As Appendix A shows, oleate tended to increase hepatic fat accumulation, but this was not significantly amplified by NMN co-infusion. Lastly, the hepatic mRNA levels of the lipogenesis gene FAS did not differ among treatments (Appendix A). OLE suppressed SREBP-1c mRNA, but this was restored by NMN + OLE, which could be attributed to hyperinsulinemia. Similarly, mRNA levels of the gluconeogenic gene Pepck were increased in both OLE and NMN, but not in NMN + OLE, also likely due to the hyperinsulinemia caused by decreased insulin clearance. The mRNA of insulin receptor and insulin degrading enzyme (IDE), which are also involved in the regulation of insulin clearance, did not differ among treatments (Appendix A).

Similar to our findings about insulin clearance, we observed a discrepancy between the present study and our previous study also with regard to β cell function in vivo (DI). Unlike our present finding in NMN-infused mice, our previous studies did not detect a decrease in DI in rats infused with the SIRT1 activator resveratrol alone (i.e., in the absence of oleate) or in saline-infused BESTO mice. Thus, we hypothesized that, similar to insulin clearance, NMN infusion causes its effect on β cell function through nicotinamide. To further investigate the effect of NMN infusion on β cells in the presence and absence of oleate, we performed immunohistochemical staining of the pancreas collected at the end of the hyperglycemic clamp. FOXO-1 staining was markedly increased by oleate (Figure 6A,B) and PDX-1 staining was inversely decreased (Figure 6C,D). NMN co-infusion normalized FOXO-1 staining and partially normalized PDX-1 staining. In the NMN-infused group, PDX-1 staining tended to be lower than control despite a lower FOXO-1 staining than control. Hence, the profile of PDX-1 staining in the four infusion groups was similar to that of DI (i.e., β cell function in vivo).

## 3. Discussion

Our hyperglycemic clamp analysis demonstrates that in C57BL/6J mice, NMN administration substantially improves glucose tolerance in the context of elevated plasma FFA levels due not only to improved β cell function, but also unexpectedly to decreased insulin clearance. This is opposite to the effect of NMN at normal plasma FFA levels, when NMN decreases glucose tolerance due to impaired β cell function. The mechanisms behind the dual effects of NMN are not clear. Nevertheless, there are several possibilities that may explain these results.

Insulin clearance occurs mainly in hepatocytes, largely mediated by CEACAM1-dependent insulin-insulin receptor complex endocytosis. Insulin clearance is known to be compromised by elevated plasma FFA levels and liver fat content [28,29,30]. Long-chain FFAs, such as oleate, are activating ligands of PPARα [31], which is highly expressed in the liver and, when activated, decreases CEACAM1 transcription [18,32]. Nevertheless, the current studies show impairment of insulin clearance by NMN + OLE co-infusion relative to individual OLE infusion despite eliciting significantly lower plasma FFA levels than in the OLE-treated group. The mechanism of the FFA reduction by co-infusing with NMN could implicate increased muscle fatty acid β-oxidation induced by PGC-1α [4,33], which is in turn activated by SIRT1-mediated deacetylation [4,34].

The substantial decrease in insulin clearance in the NMN + OLE relative to the OLE group could be attributed to the stronger lowering effect of the combination treatment on CEACAM1 expression. NMN administration increases NAD+ availability, which consequently upregulates SIRT1 activity [7,35]. NAD+ or NAD + /NADH increased in NMN infused mice and some SIRT1/ PGC-1α target genes (ACOX1 and ACO1) were upregulated. By deacetylating PGC-1α, SIRT1 could synergize with FFA-stimulated PPARα. Also, SIRT1 can activate FOXO1, which in turn can decrease CEACAM1 transcription [17]. Thus, NMN + OLE treatment could reduce CEACAM1 expression by increasing SIRT1 activity. However, we did not see any effect on insulin clearance in our previous study with the SIRT1 activator resveratrol [6]. The present findings in HepG2 cells suggest that the effect of NMN on insulin clearance is SIRT1-independent.

NMN, unlike resveratrol, can activate other SIRTs and indeed also SIRT2 can activate FOXO-1 [16]. However, Sirt2 overexpression did not decrease but rather increased CEACAM1 expression by 50% in HepG2 cells while sirtinol, which predominantly inhibits SIRT2, markedly decreased it. We hypothesized that SIRT inhibitors could be produced by NMN metabolism in the liver [36], as supported by the ~2-fold rise in plasma nicotinamide in NMN infused mice.

The mechanism whereby nicotinamide inhibition of SIRT2 could have affected insulin clearance remains to be investigated; however, sirtinol [37] and other SIRT2 inhibitors [38] exert anti-inflammatory effects, while SIRT2 may activate JNK via deacetylation [20,39]. JNK has been reported to increase [21] or decrease, CEACAM1 transcription [40]. In the present study, we could not detect any change in JNK activation in our treatment groups.

Since CEACAM1 and insulin clearance decreased significantly only in the NMN + OLE group, the effect of nicotinamide likely amplified the known inhibitory effect of FFA. Our qPCR results suggest that the possible synergism between low SIRT2 and FFA was at the level of oxidative stress, via 1) reduced insulin signaling, and/or via 2) upregulating HIF-1α/2α in the liver, which would in turn repress CEACAM1 expression in hepatocytes [24]. Unfortunately, we did not have clamp samples left to confirm impaired hepatic insulin signaling (Akt or FOXO1 phosphorylation) by Western blots in insulin stimulated conditions. We assessed serine phosphorylation of IRS1 in basal samples (as appropriate because of insulin stimulation of serine phosphorylation of IRS via MAPK and the mTOR pathway), and we could not find differences among treatments. This does not exclude the possibility of impaired insulin signaling as after 48 h fat infusion we have previously shown impairment of liver Akt phosphorylation associated with hepatic insulin resistance in the absence of changes in serine phosphorylation of IRS [22]. Regardless of its mechanism, reduced insulin clearance, in addition to substantially improving glucose tolerance, may relieve β cell overload in overcoming insulin resistance and thus prolong β cell function.

With regard to the β cell, we previously found that the SIRT1 activator resveratrol and β cell-specific overexpression of SIRT1 prevented the adverse effect of oleate on β cell function [6]. In the present study, NMN co-infusion partially prevented OLE-induced β-cell dysfunction in accordance with the previous study, although we cannot exclude that the reduction in plasma FFA levels partly contributed to the amelioration of FFA-induced β cell dysfunction in the NMN + OLE group. Notably, the Disposition Index of the NMN + OLE group could have been higher than reported had insulin secretion not been suppressed by hyperinsulinemia (caused by decreased insulin clearance).

We have previously found that inflammation plays an important role in FFA-induced β cell dysfunction [12] and we have shown that anti-inflammatory IKKβ inhibitors improve β cell function while decreasing oleate induced IRS-1 serine phosphorylation and this, possibly, upregulates the insulin signaling cascade in β cells [12]. We (unpublished) and others [41] have obtained evidence implicating β cell insulin resistance in lipotoxicity. The increased expression of FOXO-1 observed in the present study with oleate is in accordance with impairment of β cell insulin signaling induced by oleate which results in nuclear translocation and impaired cytosolic degradation of FOXO-1. The consequence is decreased expression of PDX-1 [42]. The protective effect of SIRT1 on β cell function has been in part attributed to its anti-inflammatory effect via NF-κB deacetylation [43]. Thus, it is not surprising that NMN improved β cell function in the presence of oleate in parallel with reduced FOXO-1 and associated increase in PDX-1 expression.

Intriguingly, NMN infusion in the presence of normal plasma FFA levels significantly decreased the Disposition Index, compared with the SAL group. Consistent with our results, a decrease in glucose tolerance and insulin secretion after NMN administration in male mice fed a regular diet has been reported [44]. This decrease in β cell function was not observed in our previous study with resveratrol alone or in the BESTO mouse and may therefore not be attributed to SIRT1 activation. We speculate that it could be caused by nicotinamide elevation. Whereas nicotinamide protects β cells against the adverse effect of streptozotocin [45,46], it has also been shown to impair insulin secretion [47,48]. Whether SIRT2 plays a role in β cell function and insulin secretion has not been investigated; however, it has been reported to increase glucokinase activity in the liver via glucokinase regulatory protein deacetylation [49]. Thus, glucokinase inhibition by nicotinamide in β cells could explain the decrease in DI in the NMN group. Intriguingly, PDX-1 tended to be reduced in this group relative to controls. The effect of SIRT2 on PDX-1 is not known, however SIRT1 has been reported to increase PDX-1 transcription via FOXA-2 deacetylation [50]. Thus, nicotinamide could have acted as a SIRT1 inhibitor or as a SIRT2 inhibitor if SIRT2 has the same effect as SIRT1 on FOXA-2.

Together, decreased insulin clearance and improved β cell function resulted in high insulin levels in the NMN + OLE group during the hyperglycemic clamp. The >2-fold elevation of plasma insulin levels in the clamped NMN + OLE group caused a substantial improvement in glucose tolerance although oleate-induced insulin resistance was not ameliorated. The lack of improvement of oleate-induced insulin resistance by NMN was unexpected in light of our previous observations with resveratrol [11]. Nevertheless, this discrepancy may be attributed to hyperinsulinemia as well as SIRT2 inhibition by nicotinamide [51].

Hyperinsulinemia caused by decreased insulin clearance may result in secondary insulin resistance via the downregulation of insulin receptors in hepatocytes [52]. Global and liver-specific CEACAM1 null mice, and the liver-specific dominant-negative phosphorylation-defective S503A CEACAM1 (L-SACC1) mutants develop secondary insulin resistance due to chronic hyperinsulinemia [15,26,52,53]. Hyperinsulinemia can also upregulate hepatic lipogenesis via the insulin-Akt-mTOR signaling pathway, which can contribute to hepatic steatosis and a further decrease in insulin clearance [27]. In addition, decreased hepatic CEACAM1 levels can attenuate the acute CEACAM1-mediated suppression of fatty acid synthase activity by insulin [25]. Combined, reduced hepatic CEACAM1 and hyperinsulinemia can cause hepatic steatosis. However, in the present study there was no significant amplification of hepatic steatosis in the NMN + OLE group, perhaps because this would require a more prolonged sustained hyperinsulinemic state.

This study has limitations as the ratio of C-peptide to insulin levels during hyperglycemic clamp is an index of insulin clearance that requires a relatively steady-state insulin secretion [54]. In addition, the hyperglycemic clamp is not the gold-standard method of assessing insulin sensitivity because of the non-linearity of insulin action with respect to plasma insulin levels. The hyperglycemic clamp was used in this study to investigate potential changes in β cell function.

In summary, we obtained unexpected but interesting findings that may be explained by differential effects of NMN and NMN metabolites on SIRT activation. These findings raise the question of possible untoward effects of NMN administration in humans, in light of recent findings of elevated plasma nicotinamide metabolites after oral NMN administration in healthy Japanese men [55].

In conclusion, our study provides evidence that NMN administration in the presence of elevated plasma levels of FFA, as in obesity or type 2 diabetes, results in substantially improved glucose tolerance by significantly decreasing insulin clearance and partially protecting against FFA-induced β cell dysfunction in vivo. However, in the presence of normal FFA levels, NMN administration does not affect insulin clearance and significantly reduces β cell function. These findings are of potential clinical relevance as they suggest caution in the proposed therapeutic use of NMN.

## 4. Materials and Methods

### 4.1. Animal Models

Male C57BL/6J mice obtained from Jackson Labs were housed in the Division of Comparative Medicine at the University of Toronto, under a 12 h light/dark cycle. All mice were fed a rodent diet consisting of 25% protein (% of energy), 58% carbohydrate, and 17% fat (LM-485, Harland Teklad Global Diets, Madison, WI, USA). 

### 4.2. Mouse Cannulation Surgery

All procedures were performed in accordance with the Canadian Council of Animal Care Standards and were approved by the Animal Care Committee of the Division of Comparative Medicine of the University of Toronto. Under isoflurane anesthesia, 11–12 wk old mice underwent jugular vein cannulation surgery, as previously described [12,13]. Post-surgery, cannulated mice were housed individually and provided with a recovery period of 3–5 days before intravenous infusions.

### 4.3. Intravenous Infusion of Mice with NMN and Oleate

Mice were randomized to 48 h intravenous infusions with: saline (SAL), oleate (OLE; 0.4 μmol/min), NMN + OLE (0.025 mg/kg body weight/min of NMN, based on dosage used in previous studies [7,44]), or NMN alone. OLE was infused while bound to bovine serum albumin (BSA), according to the Bezman-Tarcher method [56], at a dose that elevates plasma FFA levels by 1.5–2-fold, simulating the pathophysiologic levels present in obesity [57]. OLE and NMN solutions were freshly prepared and protected from light, as previously reported [12,13]. The pH of infusates was adjusted to 7.4 ± 0.05 prior to their administration. The infusion line was protected by a tether which allowed freedom of body movement. Mice had ad libitum access to standard chow and water. Blood samples were taken at 0 h and 46 h of infusion via the tail vein.

### 4.4. Hyperglycemic Clamp

Mice were fasted for 4 h prior to beginning the 2-h-long hyperglycemic clamp, which was performed at 46 h of intravenous infusion, as previously done [12,13]. After basal blood samples were drawn to determine glucose, FFA, insulin, and C-peptide levels, plasma glucose levels were elevated to ~22 mM (the maximum stimulatory level) via a variable intravenous glucose infusion, without interrupting the infusion of NMN and/or oleate. Glycemia was assessed every 5–10 min by measuring a drop of blood from the tail using a HemoCue Glucose 201^+^ System (HemoCue, Lake Forest, CA, USA).

After maintaining glycemia at an equilibrium of ~22 mM, blood samples were obtained to determine plasma levels of FFA, insulin, and C-peptide. Minimal blood volume was collected (0.3 mL in total/mouse) to reduce the risk of anemia, and collected red blood cells were reinfused into each mouse. At the end of the experiments, mice were euthanized via an intravenous injection of ketamine:xylazine:acepromazine (87:1.7:0.4 mg/mL, 0.2 mL/mouse). Tissues were collected, snap-frozen in liquid nitrogen, and stored at −80 °C for future analyses. Pancreas was fixed and embedded in paraffin as previously reported [58].

### 4.5. Plasma Assays

Plasma glucose levels were measured using the HemoCue Glucose 201^+^ System. Plasma insulin levels were measured using an ELISA kit (Antibody and Immunoassay Services, Li Ka Shing Faculty of Medicine, University of Hong Kong, Hong Kong) with an interassay coefficient of variation (CV) of less than 10%. Plasma C-peptide levels were measured using an ELISA kit (ALPCO Diagnostics, Salem, NH, USA) with an interassay CV also less than 10%. Plasma FFA levels were measured using a colorimetric kit (Wako and Boehringer Chemicals, Neuss, Germany). Serum nicotinamide was also measured via a colorimetric kit (Immundiagnostik AG, Stubenwald-Allee, Bensheim, Germany).

### 4.6. Immunohistochemistry

Immunohistochemistry staining was performed by the Pathology Core at The Centre for Phenogenomics. Tissue sections were submitted to heat-induced epitope retrieval with citrate buffer (pH 6.0) or with TRIS-EDTA (pH 9.0) for 7 min, followed by quenching of endogenous peroxidase with Bloxall reagent (Vector). Non-specific antibody binding was blocked with 2.5% normal horse serum (Vector), followed by incubation for 1 h in Rabbit anti-PDX-1 (Abcam, ab47267, 1:400), or Rabbit anti-FOXO-1 (Cell Signaling Technologies #2880, 1:75). After washes, sections were incubated for 30 min with ImmPRESS HRP reagent Anti-Rabbit (Vector) followed by DAB reagent, and counterstained in Mayer’s hematoxylin. Densitometric analysis was performed using Image J.

### 4.7. Western Blot Analysis

Western blot analyses were performed as previously described [14]. Liver samples were lysed in RIPA buffer containing protease and phosphatase inhibitors (Roche Diagnostics, Laval, QC, Canada). 7 µg of protein lysates were analyzed by 7% SDS-PAGE (Sodium Dodecyl Sulfate Polyacrylamide Gel Electrophoresis) gel prior to transferring proteins to a polyvinylidene fluoride (PVDF) membrane, and immunoblotting with a custom-made rabbit polyclonal primary antibody specific to α-CEACAM1 (1:500; Ab-2457), followed by horseradish peroxidase-conjugated mouse anti-rabbit IgG antibody (1:5000; LiCOR Biosciences, Lincoln, NE, USA). To assess the amount of proteins loaded on the gels, the lower half of the membrane was immunoblotted with α-tubulin antibody (1:500; Santa Cruz Biotechnology, Dallas, TX, USA). Phospho-JNK (1:300; Ab-9251 from Cell Signaling) and total JNK (1:300; Ab-9252 from Cell Signaling) and 307 ser *p* IRS1 (1:260; Ab #05-1087 from Millipore) and total IRS1 (1:100; Ab #06-286 from Millipore) and Beta-actin control (1:250; sc-47778 from Santa Cruz Biotechnology) were assessed as previously described [14]. Band densities were quantified using the ImageJ software.

### 4.8. Hepatic NAD+ and NADH Assay

The liver content of NAD+ and NADH were assayed using kit MAK037 from Sigma-Aldrich (Saint Louis, MO, USA).

### 4.9. Studies in HepG2 Cells

HepG2 cells (2 × 10^5^) were grown in 6 well plates for 24 h using MEM medium supplemented with 10% FBS, 1% L-glutamine, and 1% penicillin-streptomycin (Gibco, Thermo Fisher Scientific, Mississauga, ON, Canada). The following day, cells were treated with adenovirus-green fluorescent protein (GFP) empty vector control, adenovirus-GFP-SIRT1, or adenovirus-GFP-SIRT2 at a concentration of multiplicity of intention (MOI) 1:50 in MEM medium supplemented with 2% FBS, 1% L-glutamine, and 1% penicillin-streptomycin for 5 h before the medium was replaced with the regular growth medium. Cells were then incubated for 24 h before being starved for 48 h in phenol red-free MEM medium supplemented with 0.5% BSA, 1% L-glutamine, and 1% penicillin-streptomycin. Cells were then treated with and without 30 μM of sirtinol—SIRT inhibitor (Sigma-Aldrich) for 24 h and harvested for evaluation of SIRT1, SIRT2 and CEACAM1 expression using RT-PCR.

### 4.10. Semi-Quantitatve RT-PCR

Total RNA was isolated from livers with NucleoSpin RNA Kit (740955.50, Macherey-Nagel, Bethlehem, PA, USA). cDNA was synthesized by iScript cDNA Synthesis Kit (Bio-Rad Life Science, Hercules, CA, USA), using 1 μg of total RNA and oligodT primers. cDNA was evaluated with semi-quantitative RT-PCR in replicate (qRT-PCR; StepOne Plus, Applied Biosystems, Waltham, MA, USA), and mRNA was normalized to 18S (mice) and Gapdh (HepG2) using primers listed in Appendix A, respectively.

### 4.11. Calculations

#### 4.11.1. Insulin Sensitivity Index

When hyperglycemia is at a steady-state and is comparable between all groups examined (as in the current experimental conditions), the insulin sensitivity index (M/I = glucose metabolism/plasma insulin) can be calculated by dividing the average glucose infusion rate (GINF; normalized for body weight) by the plasma insulin concentration during the hyperglycemic clamp [59]. M/I was expressed as µmol glucose per kilogram per minute per pM insulin. Of note, this index has limitations because it is based on the assumption of linearity between GINF and the plasma concentration of insulin [60].

#### 4.11.2. Disposition Index

The Disposition Index (DI) was calculated as the product of the index of insulin sensitivity (M/I) and the plasma C-peptide concentration. Plasma levels of C-peptide, as opposed to the insulin secretion rate, were used since the kinetics of C-peptide (which are necessary for calculation of insulin secretion rate) are unknown in mice because mouse-specific C-peptide is not currently available for injection. DI has been validated by our group for use in rodents [13,61,62].

#### 4.11.3. Insulin Clearance Index

The index of endogenous insulin clearance was calculated as the ratio of C-peptide to insulin plasma levels during basal glycemia and hyperglycemia. The ratio of steady-state C-peptide to insulin plasma levels is an established index of insulin clearance [63] as insulin and C-peptide are co-secreted and C-peptide is cleared proportionally to its concentration by the kidney.

#### 4.11.4. Statistics

One-way analysis of (ANOVA) followed by Tukey’s test was used to assess significance between groups. Data are presented as means ± standard error (SE). Calculations were performed using the Statistical Analysis System software (SAS; Cary, NC, USA). *p* < 0.05 was considered statistically significant.

## Figures and Tables

**Figure 1 ijms-22-13224-f001:**
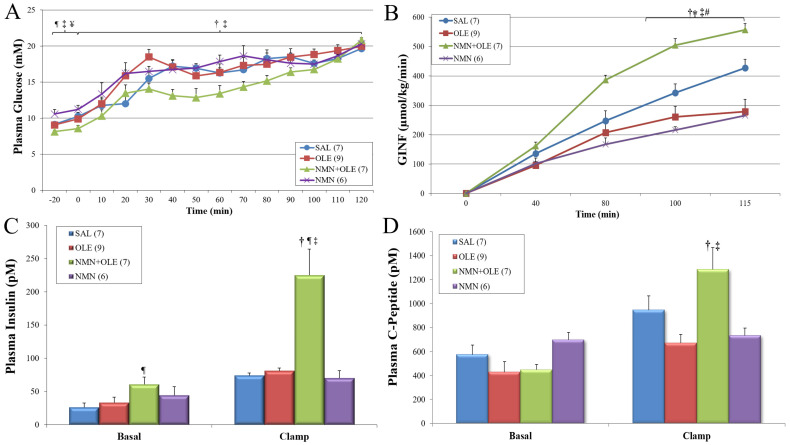
(**A**) Plasma glucose levels in mice infused intravenously with 37.5% glucose during hyperglycemic clamp. (**B**) Glucose infusion rate (GINF) required to obtain and maintain hyperglycemia. (**C**) Plasma insulin levels and (**D**) Plasma C-peptide levels prior to intravenous glucose infusion (basal period) and during the last 30 min of the clamp. C57BL/6J mice were intravenously infused for 48 h with SAL, OLE (0.4 μmol/min), NMN (0.025 mg·kg^−1^·min^−1^) + OLE, or NMN alone. These infusions were maintained during the hyperglycemic clamp. Data are means ± SE. Number in parentheses represents number of animals. * *p* < 0.05, SAL vs. OLE; ¶ *p* < 0.05, SAL vs. NMN + OLE; # *p* < 0.05, SAL vs. NMN; † *p* < 0.05, OLE vs. NMN + OLE; ‡ *p* < 0.05, NMN + OLE vs. NMN; ¥ *p* < 0.05, OLE vs. NMN.

**Figure 2 ijms-22-13224-f002:**
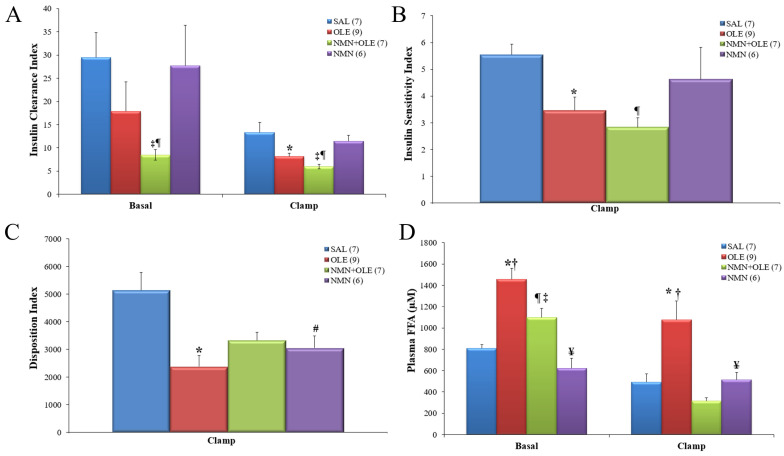
(**A**) Insulin Clearance Index (steady-state C-peptide/insulin molar ratio) prior to intravenous glucose infusion (basal period) and during the last 30 min of the clamp. (**B**) Insulin Sensitivity Index (M/I, µmol·kg^−1^·min^−1^ glucose infusion per pM insulin) and (**C**) Disposition Index (units = C-peptide in pM multiplied by SI) during the last 30 min of the clamp. (**D**) Plasma FFA levels during the basal period and during the last 30 min of the clamp. C57BL/6J mice were intravenously infused for 48 h with SAL, OLE (0.4 μmol/min), NMN (0.025 mg·kg^−1^·min^−1^) + OLE, or NMN alone. These infusions were maintained during hyperglycemic clamps. Data are means ± SE. Number in parentheses represents number of animals. * *p* < 0.05, SAL vs. OLE; ¶ *p* < 0.05, SAL vs. NMN + OLE; # *p* < 0.05, SAL vs. NMN; † *p* < 0.05, OLE vs. NMN + OLE; ‡ *p* < 0.05, NMN + OLE vs. NMN; ¥ *p* < 0.05, OLE vs. NMN.

**Figure 3 ijms-22-13224-f003:**
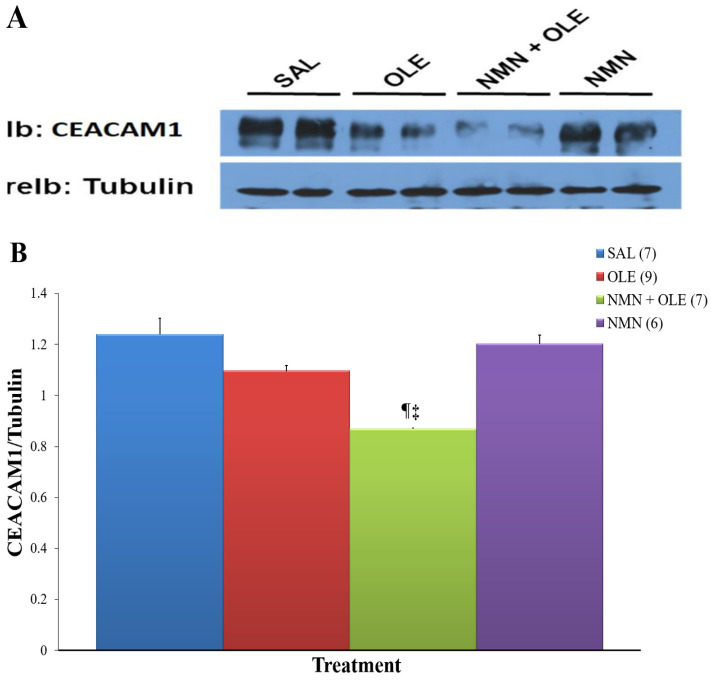
Western Blot analysis of hepatic CEACAM1 protein levels. Mice were infused for 48 h with SAL, OLE, NMN + OLE, or NMN before livers were extracted and 7 µg of protein lysates were analyzed by 7% SDS-PAGE followed by immunoblotting (Ib) the upper half of the membrane with a polyclonal antibody against α-CEACAM1 (1:500, Ab-2457) and the lower half with a monoclonal antibody against Tubulin (as a loading control). (**A**) A representative blot is shown. (**B**) Protein band density was quantified using Image J. Data are means ± SE. Number in parentheses represents number of animals. ¶ *p* < 0.05, SAL vs. NMN + OLE; ‡ *p* < 0.05, NMN + OLE vs. NMN.

**Figure 4 ijms-22-13224-f004:**
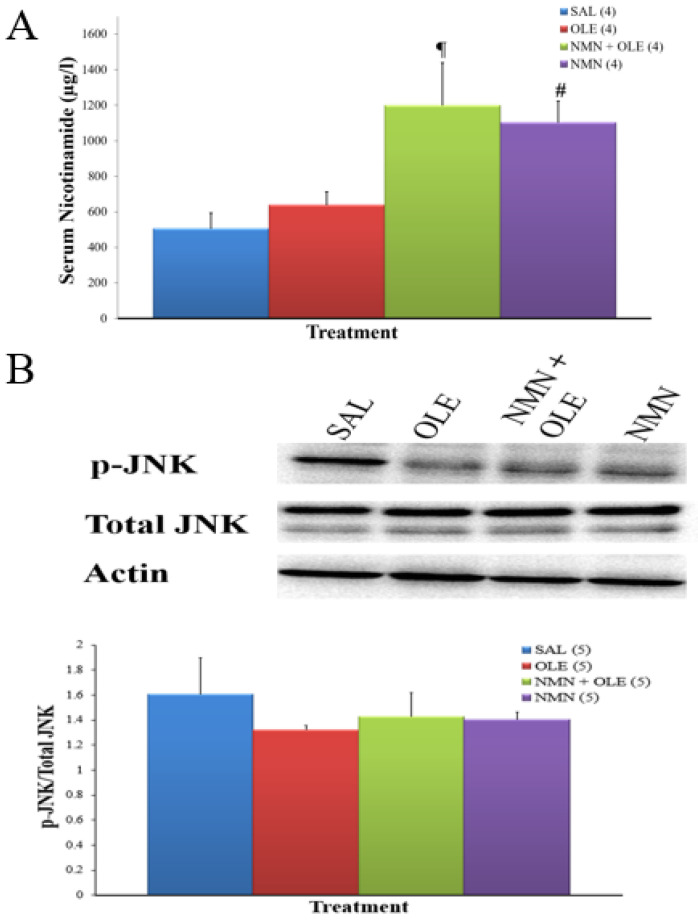
(**A**) Serum nicotinamide levels in mice after 48 h infusion. C57BL/6J mice were intravenously infused for 48 h with SAL, OLE (0.4 μmol/min), NMN (0.025 mg·kg^−1^·min^−1^) + OLE, or NMN alone. Data are means ± SE. Number in parentheses represents number of animals. ¶ *p* < 0.05, SAL vs. NMN + OLE; # *p* < 0.05, SAL vs. NMN. Western Blot analysis of: (**B**) JNK phosphorylation and (**C**) IRS serine 307 phosphorylation in the livers of mice treated as described in Figure 4a.

**Figure 5 ijms-22-13224-f005:**
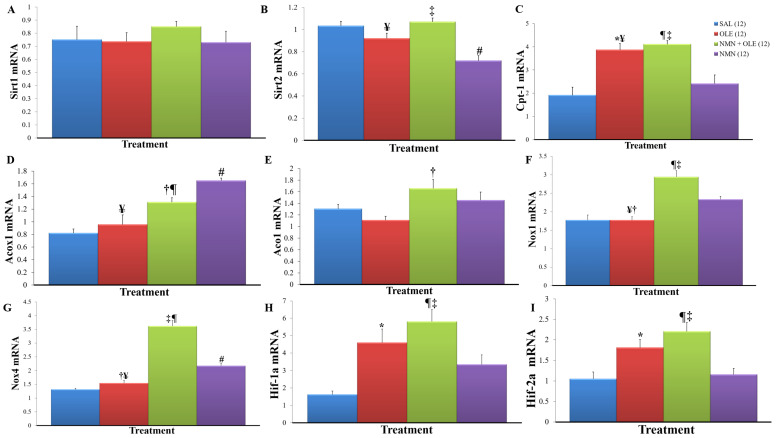
Hepatic mRNA levels after 48 h infusion. Liver lysates (*n* ≥ 6 mice/treatment) underwent qPCR analysis of mRNA in replicate and normalized against 18S. (**A**) Sirt1 mRNA levels (**B**) Sirt2 mRNA levels (**C**) Cpt-1 mRNA levels (**D**) Acox1 mRNA levels (**E**) Aco1 mRNA levels (**F**) Nox1 mRNA levels (**G**) Nox4 mRNA levels (**H**) Hif-1a mRNA levels (**I**) Hif-2a mRNA levels. Data are means ± SE. Number in parentheses represents number of animals. * *p* < 0.05, SAL vs. OLE; ¶ *p* < 0.05, SAL vs. NMN + OLE; # *p* < 0.05, SAL vs. NMN; † *p* < 0.05, OLE vs. NMN + OLE; ‡ *p* < 0.05, NMN + OLE vs. NMN; ¥ *p* < 0.05, OLE vs. NMN.

**Figure 6 ijms-22-13224-f006:**
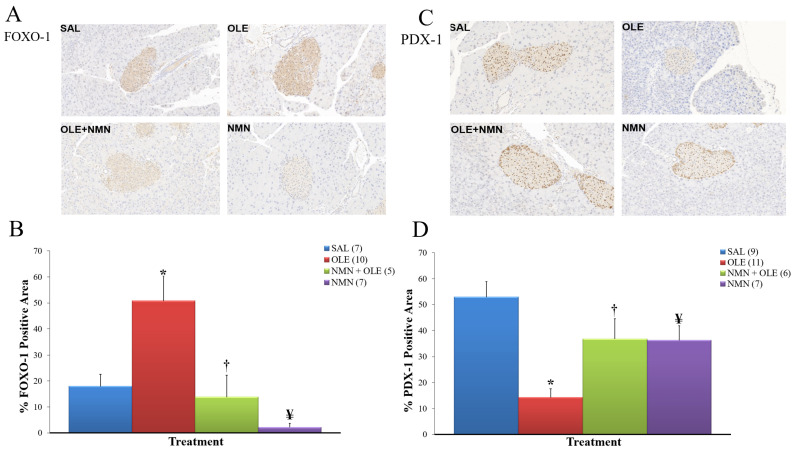
FOXO-1 and PDX-1 protein expression in pancreatic islets. Representative images of immunohistochemical staining of FOXO-1 (**A**) and PDX-1 (**C**) in pancreatic islets of mice treated with SAL, OLE, NMN + OLE, or NMN alone. (**B**,**D**) Quantification of FOXO-1 and PDX-1 positive area over total area of pancreatic islets. Data are means ± SE. Number in parentheses represents number of animals. * *p* < 0.05, SAL vs. OLE; † *p* < 0.05, OLE vs. NMN + OLE; ¥ *p* < 0.05, OLE vs. NMN.

**Table 1 ijms-22-13224-t001:** Effect of adenovirus-mediated overexpression of: (**A**) SIRT1 (**B**) SIRT2 and (**C**) SIRT inhibitor sirtinol on hepatic CEACAM1 mRNA levels in HepG2 cells. mRNA levels were assessed in triplicate/transfection and normalized to human Gapdh gene. Data are means ± SE. * *p* < 0.05 and ** *p* < 0.01.

(A)		
	Ad-Control	Ad-Sirt1
Sirt1 mRNA	<0.01	0.63 ± 0.07 **
Ceacam1 mRNA	1.66 ± 0.19	1.47 ± 0.21
**(B)**		
	**Ad-Control**	**Ad-Sirt2**
Sirt2 mRNA	1.03 ± 0.02	20.97 ± 3.62 **
Ceacam1 mRNA	0.99 ± 0.03	1.81 ± 0.07 *
**(C)**		
	**Ad-Control**	**Ad-Control + Sirtinol**
Sirt2 mRNA	1.11 ± 0.05	1.25 ± 0.20
Ceacam1 mRNA	1.02 ± 0.03	0.35 ± 0.04 **

## Data Availability

All data and other materials can be obtained from authors upon reasonable request.

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
