# Peer review of "Nicotinamide Mononucleotide Prevents Free Fatty Acid-Induced Reduction in Glucose Tolerance by Decreasing Insulin Clearance"

_ijms, 2021, doi:10.3390/ijms222413224_

Round 1

Reviewer 1 Report

Thank you very much

Author Response

We thank the reviewers for their comments. There were no further comments from Reviewer 1.

Reviewer 2 Report

The authors have provided detailed responses to most of reviewer’s comments and suggestions. New data and sentences have been included in the Results and Discussion sections according to the points raised by the reviewers. The revised manuscript is now improved. 

As minor comment, in the Results section, p. 5, line 151, the authors should replace NMN+NMN for NMN+OLE.

Author Response

We thank the reviewers for their comments. We have revised the typo identified by Reviewer 2.

This manuscript is a resubmission of an earlier submission. The following is a list of the peer review reports and author responses from that submission.

Round 1

Reviewer 1 Report

In the current work Nahle and colleagues analyzed whether the in vivo administration of NMN would prevent the deleterious effects on β-cell dysfunction and glucose tolerance induced by high plasma FFA levels in mice. To this end, NMN was injected in the absence or presence of oleate. The results show that mice with increased plasma FFA levels co-infused with NMN improved glucose tolerance, β-cell function parallel to a significant decrease in insulin clearance. The manuscript focuses on an interesting area and the conclusions are mainly consistent with the data. Nevertheless, there are some issues that need to be clarified further to support their suggestions.

 The authors assumed that 48h intravenous NMN administration in mice elevates the intracellular NAD+ levels in tissues. In fact, the authors stated in the Abstract section “NMN was infused over a 48-h period to elevate intracellular NAD+ levels and consequently increase SIRT1 activity in tissues”. Although many studies have reported that systemic NMN administration increases NAD+ biosynthesis in different tissues, including liver and pancreas, the authors should provide evidence that this is the case under their experimental conditions. Moreover, there are not results on SIRT1 activity in this work.

As commented by the authors, activation of SIRT1 and/or 2 mediated by NMN leads to FOXO-1 activation leading to the reduction of hepatic CEACAM1 mRNA and protein levels. Nevertheless, NMN administered alone has not effect on the hepatic CEACAM1 protein content compared with saline and oleate infused mice. Hence, the absence of NMN effect on hepatic CEACAM1 content should be explained. Moreover, because activation of FOXO-1 is involved in the regulation of CEACAM1 it should be of interest to show the hepatic and/or pancreas protein and phosphorylation levels of FOXO-1 in response to NMN infusion.    

Reviewer 2 Report

Dear Authors

Please see the comments below:

1-the abstract has not shown the real results. Please include hem in the abstract and just mention the main conclusion

2- the figures should be reduced or replaced by the tables showing the real numbers and p values for the authors to see the interactions

3-what is the gain of representive images of the figue 7? can be removed?

4-Figure 4 can be replaced by a table?

5-What is the main conclusion of the study? it should be the conclusion of your study without referencing.

Reviewer 3 Report

The article of Nahle and colleagues explores the combine effect of NMN and high circulating FFA in mice and their impact in b cell function and glucose tolerance.  Using hyperglycemic clamps, in mice pretreated with NMN+ oleate, the authors found and improved glucose tolerance compared to NMN alone, due to increase b cell function, an increase in glucose infusion rate and, unexpectedly, a decrease in insulin clearance. This result correlates with an impairment on CEACAM1 expression, a glycoprotein implicated in insulin receptor internalization and insulin clearance in the liver. They found that NMN effect on CEACAM1 downregulation was not due to SIRT1 activation, as they assessed SIRT1 silencing in HepG2 cells, but to SIRT2 inhibition, since NMN treatment raised serum levels of nicotinamide, a potential SIRT2 inhibitor. Finally, they explored b cell function in vivo by analyzing FOXO-1 and PDX-1 staining in the pancreas. The combination of NMN+oleate rescued both the increase of FOXO-1 and reduced PDX-1 expression in response to oleate alone. The authors conclude NMN represent a potential treatment to improve glucose homeostasis in the context of obesity and type 2 diabetes, contexts where FFA are significantly elevated. I believe the authors performed an exhaustive solid work and the results are well presented and support the conclusion. However, I found some caveats with regards to some analyses. I have several comments on these and I hope the authors will find them useful.

Major comments:

  1. Since the authors found that CEACAM1 and insulin clearance were decreased significantly only in the OLE+NMN group, it would be very informative to test whether serine phosphorylation of IRS-1 is affected and phosphorylation/activation of downstream effectors n the liver?

  1. In relation with the increased circulating insulin levels the authors found in the OLE+NMN group, I wonder if this could possibly correlate with srebp1c expression and lipogenic genes such as FAS, as well as gluconeogenic transcriptional targets of FOXO-1 (whose levels were decreased in OLE+NMN group compared to OLE alone).

  1. Related to their findings of the decrease liver insulin clearance, linked to CEACAM1 downregulation in the liver in response to OLE and OLE+NMN, did the authors assessed the expression of other key players in this process such as the INSR or IDE in these conditions in vivo?

  1. Regarding the regulation of CEACAM1 by Sirtuins by qPCR in HepG2 cells (Fig.4), could the authors please show also protein levels by WB in these same conditions?

  1. Analysis of JNK phosphorylation shown in Fig. 5 should include a protein loading control (i.e Actin, Vinculin). Is this a representative Western Blot? If so, this should be stated in the paper. Also, p-JNK in NMN+OLE group as well as NMN seem to be clearly lower than SAL or OLE group, which is not what is represented in the graph.

Minor:

  1. Line 3, please correct “symergism”.
  2. Line 225, please check English grammar in this sentence and rewrite it if necessary.